# The Neurobehavioral Phenotype of School-Aged, Very Prematurely Born Children with No Serious Neurological Sequelae: A Quality of Life Predictor

**DOI:** 10.3390/children8110943

**Published:** 2021-10-20

**Authors:** Barthélémy Tosello, Sahra Méziane, Noémie Resseguier, Stéphane Marret, Gilles Cambonie, Meriem Zahed, Véronique Brévaut-Malaty, Any Beltran Anzola, Catherine Gire

**Affiliations:** 1Department of Neonatology, North Hospital, University Hospital of Marseille, Chemin des Bourrely, CEDEX 20, 13915 Marseille, France; Sahra.MEZIANE@ap-hm.fr (S.M.); meriem.zahed@ap-hm.fr (M.Z.); veronique.brevaut@ap-hm.fr (V.B.-M.); any-alejandra.BELTRAN-ANZOLA@univ-amu.fr (A.B.A.); catherine.gire@a-hm.fr (C.G.); 2Aix Marseille Université, CNRS, EFS, ADES, 13915 Marseille, France; 3CEReSS-Health Service Research and Quality of Life Center, Faculty of Medicine, Aix-Marseille University, 27 Boulevard Jean Moulin, 13005 Marseille, France; noemie.resseguier@ap-hm.fr; 4Department of Neonatal Medicine, Neuropediatrics Rouen University Hospital and INSERM U 1245, Neovasc Team, Perinatal Neurological Handicap and Neuroprotection IRIB, School of Medicine, Rouen University, 1 rue de Germont, CEDEX, 76031 Rouen, France; Stephane.Marret@chu-rouen.fr; 5Department of Neonatal Medicine, Montpellier University Hospital, 191 av. du Doyen Giraud, CEDEX 5, 34295 Montpellier, France; g-cambonie@chu-montpellier.fr

**Keywords:** extremely pre-term children, neurocognitive/behavioral disorders, quality of life, anxiety

## Abstract

School-aged extremely preterm (EPT) children have multiple specific neurocognitive/behavioral disorders that are often associated with other disorders; this manifests a true neurobehavioral “phenotype” of prematurity. To determine a profile of cognitive/behavioral impairments in a population of school-aged EPT children (7–10 years-old) without major disabilities, a cross-sectional study was conducted in five medical centers. An algorithm distributed the study population according to four WISC-IV subtests, five NEPSY-2 subtests, and two variables of figure of Rey. The behavior (SDQ), anxiety (Spielberg STAI-C), and generic QoL (Kidscreen 10 and VSP-A) were also evaluated. The study included 231 school-aged EPT children. Three neurobehavioral “phenotypes” were defined according to their severity: 1 = moderately, 2 = minor, and 3 = unimpaired. In all the profiles, the working memory, perceptual reasoning, as well as mental flexibility, were close to or below average, and their emotional behavior was always troubled. Self-esteem and school-work were the most impacted QoL areas. The unimpaired neurobehavior exhibited emotional behavioral impairment and executive dysfunction. The profile analysis defined distinct outcome groups and provided an informative means of identifying factors related to developmental outcomes. The QoL deterioration is determined by the severity of the three neurobehavioral “phenotypes”, which is defined as well as by dysexecutive and/or behavioral disorders.

## 1. Introduction

There is a remarkable consistency in outcomes over time and between countries, cultures, and healthcare systems, which provides evidence for a universal “preterm phenotype” associated with the neurodevelopmental immaturity conferred by a VP birth [1]. Relative to those born at term, very preterm (VP) children are at an increased risk for cognitive impairments, attention deficits, and social–emotional problems. However, there is no increased risk for disruptive or oppositional behavioral problems [2]. Moreover, developmental coordination disorders (DCD) are associated with behavioral, cognitive, and/or attention disorders [3,4] in EPT school-aged children who are free from severe deficiencies. In addition, two meta-analyses confirm a high prevalence of psychiatric disorders in children and adults born prematurely as compared to a population born at term. These VP children have a high risk of attention disorders with or without hyperactivity, autism spectrum, and anxiety disorders [1,5]. However, these disorders are always accompanied by cognitive, neurological, and learning comorbidities [6].

As traditionally seen in the literature in cohort populations, the cognitive fate of premature children is usually classified into four groups: (1) absence of incapacity, (2) minor disability, (3) moderate disability, and (4) severe disability, based on data from the total Full Scale Intelligent Quotient (FSIQ) scores. A threshold of ≥89 is considered normal, which is a low average [4,7]. However, the performances observed with a total FSIQ are the result of a complex processes involving multiple intellectual and non-intellectual characteristics (attention, emotions, motivation, movement planning, etc.) [8]. Furthermore, for infants born prematurely, the FSIQ is most often calculated on a dissociated subtest value that is not necessarily reflective of the child’s cognitive function and thus constitutes a false methodology [8]. The analysis of the subtests’ dispersions aims to highlight those that deserve a thorough interpretation. Indeed, a child considered “normal” is likely to have impairments such as a dysexecutive syndrome and/or impaired behavior and/or DCD that could disrupt brain function.

Therefore, most longitudinal studies focus on general mental health classifications rather than individual clinical diagnostic criteria and comorbidity profiles (motor and behavior). Since some outcomes may be deteriorating rather than improving over time, knowledge of all comorbidities may be useful in improving an early identification of the highest risk children in order to address their clinical needs. Inclusion of behavioral difficulties and developmental coordination disorders with the definition of mild neurodevelopmental disabilities describes, in more depth, the complexity of those difficulties faced by these preterm children [9].

In fact, there have been few studies directed on individual clinical descriptions in which discrete or non-specific signs of psychiatric disorders and/or subtle impairments of executive functions occur together and/or are associated with cognitive and/or neurological impairments in school-aged VP children. To our knowledge, only one study of these children who share similar profiles within a group on measurements of intelligence and executive functions does not take into account behavior [8]. The assessment of a true neurobehavioral “phenotype” (NBP) of prematurity, which takes into account not only the FSIQ but also executive functions, neurologic examinations, and behavior should be a principal objective of epidemiological research on prematurity [8]. Finally, no studies have measured this neurobehavioral profile and its social consequences particularly on the QoL of these children.

The objective of our study was to identify neurobehavioral subgroups in an EPT population who were free from those severe disabilities included in the GPQoL study: (1) shared similar profiles on the measurements of intelligence and executive functions, (2) the distribution of anxiety and behavioral disorders assessment in each group, and (3) a correlation with QoL measurements

## 2. Method

### 2.1. Study Design and Population

Our population was obtained from the sample included in the GPQoL study of EPTs at school-age with no resultant major disabilities. The GPQoL study is a cross-sectional, multi-center cohort investigation of school-aged EPT children (<28 weeks gestational age (GA)) who were discharged from the hospital alive and free from severe disabilities such as severe cerebral palsy, autism, and/or severe intellectual disabilities (FSIQ < 65). This study was conducted within five French Level III facilities authorized to care for preterm infants less than 28 GA (Marseille Conception, Marseille Nord, Nantes, Nîmes, and Rouen). Our participating centers care for 20% of EPTs born in France each year. A detailed description of the GPQoL study was published previously [10,11]. All patients included had a long-term evaluation at school age (between 7 and 10 years old) including a clinical examination, along with their cognitive and behavior functions [10,11].

### 2.2. Measurements Collected from the GPQOL Study Data

#### 2.2.1. Clinical and Sociodemographic Data

Pregnancy and perinatal data were obtained from the patients’ medical files. The sociodemographic and family data were collected at their inclusion to the GPQoL study [10,11]. This information included age, gender, parental education, parents’ professional activity, and the child’s school life. The Family Affluence Scale (FAS) assessed the family’s socioeconomic status from the child’s perspective.

#### 2.2.2. Psychometric and Behavioral Evaluation Data (7–10 Years Old)

Motor skills were assessed by the Touwen Infant Neurological Examination (TINE). The psychometric tests were carried out, using the WISC-IV (Weschler Intelligence Scale-for Children—4th edition) and its four indices to calculate the FSIQ: (1) Verbal Comprehension Index (VCI), (2) Perceptual Reasoning Index (PRI), (3) Working Memory Index (WMI), and (4) Processing Speed Index (PSI) (mean 100 and standard deviation (SD) 15); the figure of Rey: a short memorization and perceptual organization test, and the NEPSY-2 (NEuroPSYchological Assessment—2nd edition), which included subtests evaluating attention and executive functions: planning (tower), mental flexibility (fluidity of design), and cognitive inhibition (statue) (means 10 and standard deviation 3). Our psychologists received special training so as to ensure homogeneity in their evaluations. Finally, behavioral assessment was carried out by the questionnaire for parents: SDQ (Goodman Strengths and Difficulties Questionnaire). The assessment of anxiety was evaluated by the Spielberg STAI-C (State–Trait Anxiety Inventory) questionnaire for children.

#### 2.2.3. Quality of Life Data

Data were obtained by using two general QoL questionnaires. A self-assessment was used for children and a hetero-assessment questionnaire for the parents: Kidscreen-10 score and VSPA (*Vécu et Santé Perçue des Adolescents*). VSP-A is a questionnaire with a total index measuring nine dimensions: vitality, psychological well-being, relationships with friends, hobbies, relationships with family, physical well-being, relationships with teachers, school-work, and self-esteem. These data were obtained using the children’s VSP-Ae versions and the adult’s VSPA-p versions. The Kidscreen’s full version questionnaire explored the following dimensions: physical well-being, psychological well-being (positive and negative), emotions, parental relationships including independence from their parents, relationships with friends, social relationships, and school work. The GPQoL study used the 10 items of the short children’s and parents’ versions (Kidscreen-10) to obtain a total index. The data from the QoL scores were compared to the 2003 European database’s reference population’s data [10,11]: a French sample (*n* = 989) obtained from randomized telephone calls (CATI method: Computer-Assisted Telephone Interview; RDD: Random Digital Dialing).

#### 2.2.4. Disability and Specific Cognitive Impairment Definition in the GPQoL Study

Disability in the GPQoL study was based on the FSIQ score [10,11]. A specific cognitive impairment was considered in the GPQoL study [10,11] based on classifications of mental illnesses according to the Diagnostic and Statistical Manual of Mental Disorders, fourth edition (DSM-IV (footnotes Table 1)).

### 2.3. Ethics

This study has been approved by the Personal Protection Committee (18 December 2012, reference 12.018) and is registered on ClinicalTrials.gov-NCT01675726.

### 2.4. Statistical Analysis

#### 2.4.1. Descriptive Analysis 

A descriptive analysis was first carried out with qualitative variables presented as numbers and percentages and quantitative variables presented as means and SD. For quantitative variables, the assumption of a normal distribution was checked using histograms and plots (QQ methods (quantile–quantile)). The study’s sample size was sufficient to use a Gaussian distribution and apply parametric procedures even if the distributions were somewhat far from normal. Sample analysis data who had all the psychometric tests of WICS-IV and NEPSY-2, behavioral assessment, and QoL examinations and with no missing data were compared between GPQoL participants [10,11].

#### 2.4.2. New Classification of Cognitive/Behavioral Impairments

To establish the new “classification” of neurocognitive/behavioral impairments in EPT children, clusters used all the psychometric evaluations of the GPQoL study. The following data were considered: four subtests of WISC-IV (VCI/PRI/WMI/PSI), five subtests of NEPSY-2 (statue, tower, fluidity of pattern, selective auditory, visual attention), and two variables of the figure of Rey (copy and reproduction). We used the PAM algorithm (Partitioning Around Medoids) [12,13] with the only constraints being the number of clusters to be established. This number was defined a priori on clinical assumptions. The algorithm sought a distribution to obtain the maximum similarities within each cluster as well as the maximum differences between the clusters. For the clustering, the distances between the observations were calculated with the Euclidean method (square root of the sum of the squared deviations). Then, all data were compared according to the three established neurobehavioral profiles for qualitative data, the χ^2^ test was used when valid (otherwise, the Fisher test was used). For quantitative data, the analysis of variance was used when valid (otherwise, the Kruskal–Wallis test was used).

#### 2.4.3. QoL Analysis

Comparisons of QoL scores between the study population and the reference population (sex and age-matched) were performed using a paired *t*-test. Observed differences and their 95% Confidence Intervals (CI) were estimated, as were the effect sizes using the Cohen d statistics for paired data to rank the dimensions of the VSP-A in a descending order of difference. The same analysis was performed in each established neurobehavioral profile (data only shown for the less impacted profile). As suggested by Cohen, d statistics lower than 0.2 represent a negligible difference even it is statistically significant while d statistics equal to 0.2, 0.5, and 0.8 represent small, medium, and large differences, respectively.

Statistical analyses were performed using R-software, version 3.6.0. All tests were two-sided. The statistical significance was defined as *p* < 0.05.

## 3. Results

### 3.1. Population

We categorized 231 children into three clusters (Figure 1) with a mean age of 8.46 (±0.75) years. The FSIQ mean was 91.62 (14.93) (low mean average), and 106 (59.55%) had a dysexecutive syndrome. The mean emotional symptoms were slightly elevated (3.48, 2.42). The comparison between children who had all examinations (psychometric tests of WISC-IV and NEPSY-2, behavioral assessment, and QoL) and those children who had at least one missing data of these examinations did not show any statistical differences (Appendix A).

### 3.2. Cognitive/Behavioral Impairments Profiles from New Classification

The 231 children were classified according to the severity of their neurobehavioral assessment into three neurobehavioral impairment groups:
Cluster 1 Moderate Impaired NeuroBehavior (MODINB) with multiple impairments and behavior troubles (*n* = 46 (20%));Cluster 2 Minor Impaired NeuroBehavior (MINB) with less severe multiple impairments and behavioral troubles, (*n* = 99 (43%)); andCluster 3 (*n* = 86 (37%)), UnImpaired NeuroBehavior (UINB), with only slight emotional symptoms (mean score: 3.5 (SDs) = 2.56: slightly below average, therefore pathological), with a normal neurocognitive assessment outcome.

Behavior and anxiety were significantly different between each different cluster and with a gradient of severity (Table 1).

### 3.3. Neurobehavioral Impairment Profiles: Behavioral and Neurocognitive Comorbidities

In Cluster 1 (MODINB), emotional troubles had high scores (4.05 (2.62)), and hyperactivity was slightly below the mean (5.32 (2.45)). The anxiety score was (35.30 (8.28)), 30 children (65.22%) had language delays, and 44 (95.65%) had a dysexecutive syndrome. The other impairments (visuo-spatial integration delay, attention deficit disorder, and ideomotor dyspraxia) were presents in 15 to 17% of the cases (Table 1).

In Cluster 3 (UINB), emotional symptoms were always below the mean and only 26 (33%) of the children had a dysexecutive syndrome. Dysexecutive disorders were a composite of performance across four subtests: working memory index (WMI) < 85 and/or planning score < 8 and/or the mental flexibility score < 8 and/or inhibition score < 10th. It may be that some children had slightly lower performances (just less than −1 SD) on just one of these tests over −1 SD (Table 1).

### 3.4. Neurobehavioral Cognitive Assessment by Cluster

In Cluster 1 (MODINB), the WMI and the PRI were close to −2 SDs, 74.93 (SD = 11.07), and 77.26 (SD = 8.99) respectively, the other indices were close to −1 SD (Table 1 and Figure 2).

In Cluster 2 (MINB), the PSI and the PRI were close to −1 SD, respectively 88.49 (SD = 10.75) and 85.41(SD = 9.24); the other indices (WMI and VCI) were close to the low average.

In Cluster 3 (UINB), the WMI was close to the average as was the PSI, respectively 101.15 (SD = 11.41) and 102.90 (SD = 12.61). The other subtests VCI and PRI were above average with a difference of 0.5 SDs.

Mental flexibility was below average (9.48 (SD = 2.68)) in Cluster 3 (UINB), less than −1 SDs (7.76 (SD = 2.72)) in Cluster 2 (MINB), and less than −2 SDs (6.50 (SD = 2.27)) in Cluster 1 (MODINB) (Table 1 and Figure 3).

All clusters had a behavioral score slightly below average for the variable “emotional symptoms”. Cluster 2 (MINB) had a slightly below average “social relationship disorder” score. Cluster 1 (MODINB) had a slightly below average scores for all variables in the Goodman test, except for the variable “prosocial behavior” (Figure 4).

### 3.5. Correlation of Population Study and Quality of Life

The QoL was reduced for all children when compared to the reference population. The exceptions to this were the relationships with the family and the teacher, as evaluated by the parents. For the children, the areas impacted were ranked as (1) Self-esteem, (2) Relationship with friends, and (3) Hobbies. The parents cited the areas most impacted for their children were (1) Psychological well-being, (2) School-work, and (3) General well-being (Table 2).

#### 3.5.1. Quality of Life Comparisons between the Neurobehavior Impairment Clusters

There are significant QoL differences between the clusters (as indicated by the total VSP-A index), in self- and hetero-evaluations. The QoL deterioration is determined by the severity of the three neurobehavioral “phenotypes”, which are defined as well as by dysexecutive and/or behavioral disorders. The areas most specifically concerned were the school-work and self-esteem categories (Table 2).

#### 3.5.2. Quality of Life in Cluster 3 

UINB as compared to the reference population (Table 2). There was a significant reduction in the QoL in Cluster 3, as compared to the reference population. The most impacted areas for the children were (1) Relationships with friends, (2) Leisure, and (3) General well-being. From a parental point of view, the areas most affecting their children were (1) Psychological well-being, (2) General well-being, and (3) Vitality. The area of school-work was not impacted.

## 4. Discussion

We defined broader neurobehavioral profiles, in an individual classification, using three groups of children with different severities from school-aged EPT children with no severe disabilities. There were two group of children with multiple cognitive and behavioral impairments with a gradient of severity (Moderate (MODINB) and Minor (MINB)) and one group of children was classified with an unimpaired cognitive outcome with only slight dysexecutive impairment and emotional behavioral symptoms (UINB). There was a reduction in the QoL in the three NBPs as compared to the reference population confirming the relevance of clustering. The QoL areas most impacted, from the children’s perspective, were self-esteem, relationship with friends, and hobbies. From the parent’s perspectives, the areas most impacted were psychological well-being, school-work, and general well-being. This is consistent with our original study [10]. However, the more the NBP was altered, the more the QoL was decreased. Eventually, the decrease in QoL in an “unimpaired” NBP could be explained by the inability of the VP child to manage his/her emotions and/or executive dysfunction [11].

The Epipage 2 study recently included an analysis of full-scale intelligent quotient (FSIQ), total behavioral difficulties, and developmental coordination disorders to obtain a “composite” score to evaluate neurodevelopmental disorders [9]. This confirmed that an evaluation of the long-term fate of children born preterm is challenging not only because the completeness of even minor disorders is difficult to specify individually but also because the outcome measures must be meaningful at the time, for health professionals, policymakers, parents, and children. Without taking into account the behavioral analysis, the Heeren study [8] classified four subgroups of the neurocognitive profile in the Elgan cohort: one normal, one with a low normal profile, where the impairment was mainly of the executive functions, and two with diffuse impairment (cognitive and executive functions) observed for moderate and severe profiles. This study showed that the FSIQ was insufficient to characterize moderate or minor cognitive impairment of the EPT and that the impact of the executive disorders, such as inhibition, working memory, and mental flexibility, was minimized. In our study for Cluster 3, classified as “unimpaired”, the mean FSIQ score ranged from 101 to 109, and the NePSY subtest score ranged from 9.5 to 11.6. These scores are all within normal range. Only the associated dysexecutive disorders are elevated (30%), and this is a composite of performances across three subtests: Working Memory Index (WISCI-IV), planning, and mental flexibility (NePSY-2). It may be that these children had low performance (just less than -1 ds) on just one of these tests. These findings confirm the role of the worsening of impaired executive functions on the worsening of the cognitive profile of former premature babies and the importance of taking them into account in the description of the clinical outcome [8].

However, a prospective study [3] on a population sample of school-age EPTs with no serious sequelae showed that DCDs were most often associated with comorbidities such as behavioral and/or executive and/or attention disorders. On the other hand, behavioral disorders were present in an identical manner in premature children with or without DCD. This justified long-term follow-up with a complete behavioral, motor, and neurocognitive measurement and thus neurobehavioral profile such as in our work. 

Mathewson et al. performed a metanalysis of the behavioral profiles of very premature babies during the school- aged years and in adolescence versus controls born at term [5]. Selecting a population of 2004 premature babies versus 1238 controls shows that the behavioral profiles, depending on their severity, are specific and associated with cognitive or neurological comorbidities. As in our study, in the event of an NBP impairment with minor and moderate difficulties, the behavioral score and the cognitive function were altered. In the NBP with only a dysexecutive disorder, only emotional behavioral disorders were highlighted. Johnson et al. [1,14] proposed, for the first time, the concept of “behavioral phenotype of premature babies”, which would have an impact during their school-aged years and would be characterized by attention deficit/hyperactivity disorders, social and emotional difficulties, and introversion [15,16]. It was the first systematic investigation to disentangle the proposed comorbidity of the preterm behavioral phenotype suggesting, as it did in our study, that there may be more than one behavioral and/or cognitive profile seen in outcome preterm births, some of which may be common to outcome full-term births. Burnett et al. [17,18] confirm this association with a worsening of behavioral disorders, which was also seen in our study, in association with neurological comorbidities, using the theme of cluster-based approaches similar to our study, and they identified four behavioral profiles in five-year-old children born very preterm: Profile 1 grouped those typically developing children who exhibited a neurodevelopmental and psychiatric level similar to that in the general pediatric population. Profile 2 grouped at-risk children with lower neurodevelopmental scores and slightly elevated psychiatric profiles that remained within the typical range. Profile 3 represented the psychiatric group that included children with moderately severe to severe difficulties with executive function and attention deficit hyperactivity disorder (ADHD) and autism spectrum disorders (ASD) symptoms as seen by both the parent and the teacher. Profile 4 characterized children into an inattentive/hyperactive group classified by low cognitive and low language scores, which was reported by many parents and teachers as ADHD. Finally, Korzeniewrski et al. [19] found that children with an IQ greater than 85 had social adjustment disorders in 16% of the cases associated with attention deficit, executive disorders, language, communication, and emotional disorders. This association of disorders is even more important when the FSIQ is <85, as found in our Clusters 1 and 2.

Our work shows that the most affected cognitive areas were similar between the different neurobehavioral profiles: PSI, WMI, PRI, mental flexibility, and emotional behavior with a significant worsening depending on the severity of the NBP. The literature [20,21,22,23] reports that the PSI and WM are independent predictors of academic difficulties amongst the very premature-birth preschool children. Other authors [2,24] hypothesize that a deficit in WM and/or attention and/or PSI, which impact other mental processes, would be the cause of later deficiencies such as speech delay or dysexecutive disorders. According to the author, the PSI deficit (which is dependent on GA) is correlated to executive functions: WM (verbal and visuo-spatial), inhibition, and cognitive flexibility [20,22,25]. Similarly, as in our study, there is a correlation between the measurement of the PSI and/or WM and the behavioral symptoms of impulsivity/hyperactivity and attention disorders.

Our study shows that the NBP of prematurity is a diffuse disorder with a defect of multiple “functions” such as cognitive (executive function), motor, and behavioral disorders, which influences social adaptation. Social adjustments can be indirectly measured by the quality of life (QoL) evaluations. QoL is an individual’s subjective perception of his state of health measured by means of his basic needs: biological, human relationships, work, and leisure [10]. Recently, the QoL of those school-aged EPT children born in France without serious sequelae has been reduced as compared to the reference population [11]. The independent determinants of this population’s QoL are language comprehension disorders, visuo-spatial disorders, executive disorders, and behavioral disorders [11]. These outcomes when the children reach school-age are indicative of their life-long functioning.

A systematic review [26], using instruments such as QoL and measurements of social adaptation and behavior, showed that EPTs had poor social adaptation skills, which appeared early and were persistent during childhood and adulthood, with its severity depending on the GA, brain abnormalities, and socioeconomic status. In our cohort, there was no correlation between the intensity of the neurobehavioral impairment and the GA [20] but rather a correlation with a low weight for GA. There are hypotheses that suggest that this NBP of prematurity is due to cerebral hypo-connectivity, thus leading to diffuse structural anomalies that may sometimes be similar to autism spectrum disorders [27] (although there is a huge spectrum) but with a reduced severity. Connectivity profiles associated with preterm births have been studied in the context of several different psychiatric disorders [1]. The mechanism of cerebral hypo-connectivity in EPTs is correlated with the existence of attention deficit disorders in childhood with one of the predictors being a low growth weight [28].

Our study’s main limitation was the absence of a control group. As in our previous study [11], we chose a European sample to compare our QoL study population to a reference population. The cluster classification of our population studied represented 70% of those initially analyzed in the GPQoL, but our population study was a sample, which is representative of a national French sample without serious sequelae. For correlations, we only analyzed 77% of the 231 patients in the eligible sample, but there are no missing data. 

Our study’s strengths included a homogeneous, multicentric, large population, representative of a national sample. There were no differences in perinatal, neonatal, socioeconomic, and QoL characteristics from the included population against the non-included. The psychometric properties of our two QoL questionnaires were validated on a large sample of children. Our study used the classification in the NBP according to severity in order to describe a population of EPT school-aged children free from serious neurological sequelae. The neurobehavioral profile in sub-groups of severity, with below-average performance rather than isolated specific cognitive and/or behavioral and/or motor impairments, appears to be more representative of reality. This has relevance, since there is an inverse correlation with QoL. It is important to note that the areas most affected in our study, in the case of sub-normal clusters, are self-esteem and mental well-being.

The prevalence of neurobehavioral disorders currently dominates the EPT’s state at school age. Its persistence and worsening in adulthood with introverted personality and anxiety disorders leads to withdrawal and social distancing, which makes it a public health problem [26]. Knowing the long-term future of EPTs is paramount [29] as it provides scientifically based, reliable information to parents and improves care by promoting the prevention of these neurobehavioral disorders [30]. It is for this reason that an early pre-school-age determination of a neurobehavioral phenotype may allow more focused surveillance and/or development of intervention strategies [31,32]. Cluster analysis defined distinct outcome groups in EPT children and provides an informative means of identifying factors related to developmental outcome. Finally, socio-behavioral and emotional capacities are associated with executive functions and educational pathways [33]. Tools for “re-educating” executive functions are suggested and could be integrated into school programs [34].

## Figures and Tables

**Figure 1 children-08-00943-f001:**
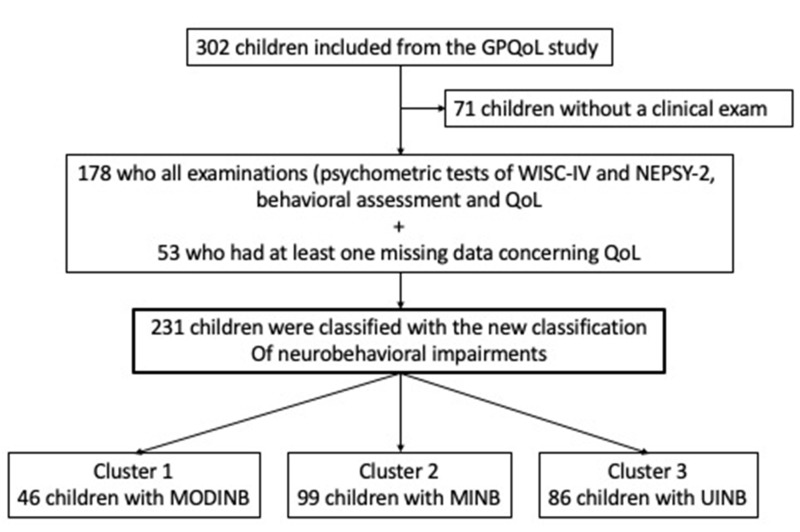
Study population’s flow chart. Note: Quality of Life (QoL); Cluster 1: Moderate Impaired NeuroBehavior (MODINB); Cluster 2: Minor Impaired NeuroBehavior (MINB); Cluster 3: UnImpaired NeuroBehavior (UINB).

**Figure 2 children-08-00943-f002:**
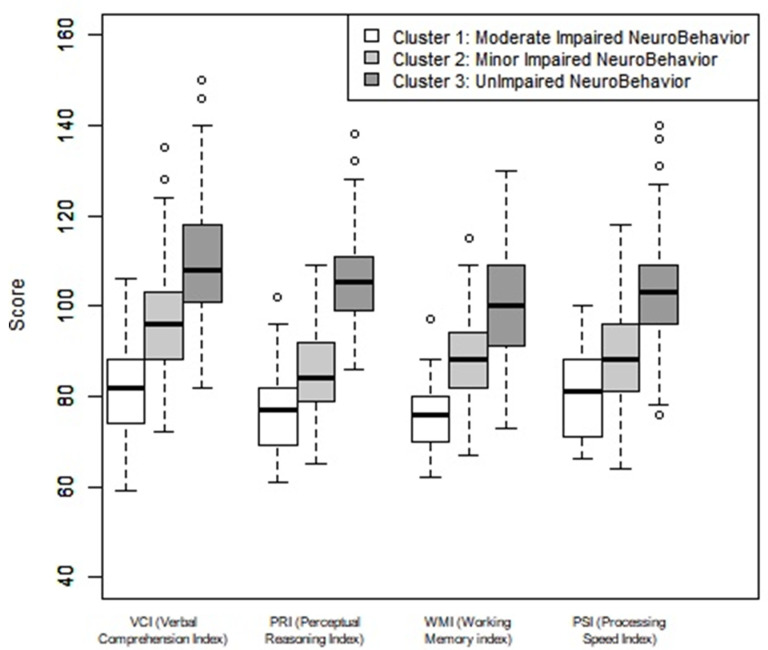
Neuropsychological evaluation of the four WISC IV indices by cluster. Note: Verbal comprehension index (VCI), perceptual reasoning index (PRI), working memory (WMI), and processing speed index (PSI). Average = 100, 1 SD = 15.

**Figure 3 children-08-00943-f003:**
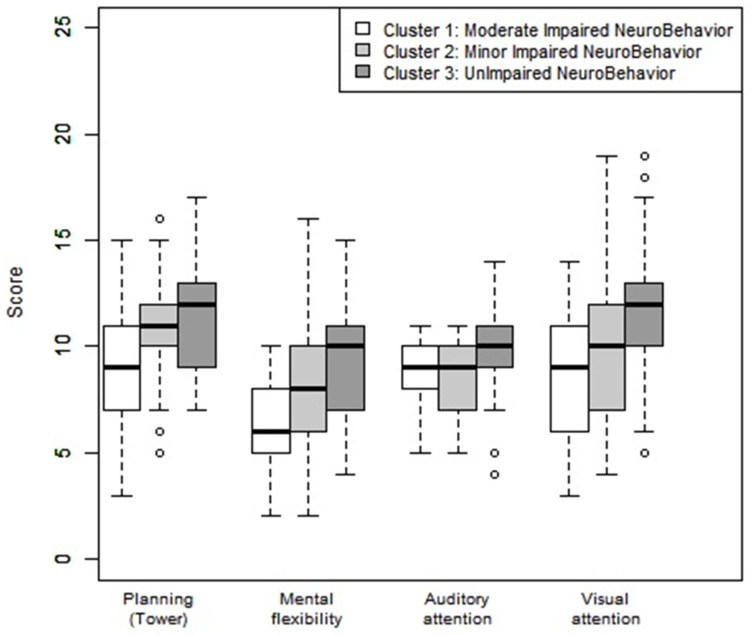
Neuropsychological assessment: planning, mental flexibility, auditory, and visual attention according to the clusters. Note: Average: 10, deviation: 3. A <8 score reflected a disorder in the category concerned: <−1 SD).

**Figure 4 children-08-00943-f004:**
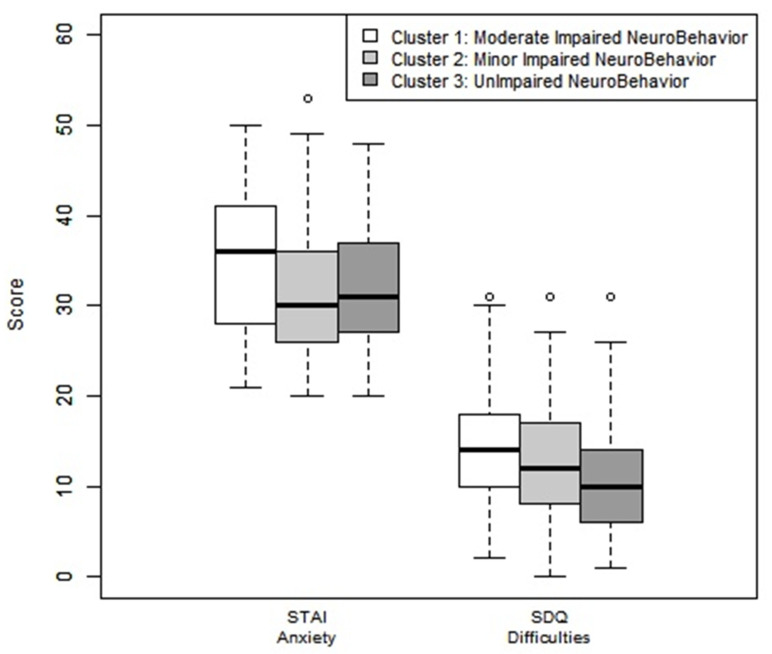
Anxiety assessment by STAI questionnaire and behavior assessments by Goodman-SDQ test by clusters. Note: Goodman-SDQ (Strengths and Difficulties Questionnaire): Scores were correlated with Achenbach’s Child Behavior Checklist (CBCL), including 25 items targeted for the parents. The questionnaire reflected a total assessment score of the difficulties and defined five subcategories composed of five items each: emotional disorders: average: 0–3, slightly below average: 4, high: 5–6, very high 7–10; *behaviors: average*: 0–2, slightly below average: 3, high: 4–5, very high 6–10; hyperactivity: average: 0–5, slightly below average: 6–7, high: 4–5, very high 6–10; *social relationship disorder: average*: 0–2, slightly below average: 3, high: 4–6, very high 7–10. A *total difficulty score* was the sum of the first four items, with each one-point increase corresponding to an increase in the risk of developing a mental disorder. The categories assessed was in the average 0–13, slightly below the average 14–16, high 17–19, or very high 20–40. “Prosocial” behavior is counted separately and varies in the opposite direction: average: 8–10, slightly below average: 7, high: 6, very high 0–5. STAIC: State–Trait Anxiety Inventory for children. Schematically: standard anxiety state: Normal: 20–40, High Anxiety: higher than 40.

**Table 1 children-08-00943-t001:** Characteristics of the clustered pooling population.

			Characteristics of Clustered Pooling (N = 231)			
			MODINB Cluster 1 (N = 46)	MINB Cluster 2 (N = 99)	UINB Cluster 3 (N = 86)	*P*
**Perinatal characteristics**						
	GA in WA, mean (SD)		26.13 (0.96)	26.25 (0.82)	26.29 (0.92)	0.607
	Weight in grams, mean (SD)		858.91 (181.77)	878.77 (194.69)	901.31 (171.47)	0.430
	Male, n (%)		20 (43.48)	50 (51.02)	38 (44.71)	0.594
	SGA, n (%)		5 (10.87)	10 (10.10)	1 (1.16)	0.011 *
	Multiple pregnancies, n (%)		17 (36.96)	30 (30.30)	31 (36.05)	0.625
	Severe BPD, n (%)		25 (54.35)	46 (47.92)	47 (54.65)	0.613
	Average age of child at the study inclusion, mean (SD)		8.69 (0.65)	8.43 (0.74)	8.39 (0.72)	0.056
**Parents’ educational level, professional activity, and socioeconomic status of family**						
	Parents without higher education level, n (%)		29 (74.36)	46 (47.92)	18 (21.43)	<0.001 *
	Professional activity of parents, n (%)					0.004 *
		Without professional activity	8 (18.60)	7 (7.14)	3 (3.53)	
		Professional activity of one of two parents	16 (37.21)	37 (37.76)	20 (23.53)	
		Professional activity of both parents	19 (44.19)	54 (55.10)	62 (72.94)	
	Professional activity of mother, n (%)		21 (48.84)	63 (64.29)	66 (79.52)	0.002
	Professional activity of father, n (%)		33 (82.50)	82 (89.13)	78 (92.86)	0.243
	Elevated FAS Score, n (%)		21 (45.65)	57 (58.76)	57 (67.06)	0.059
**Quality of life ^1^, mean (SD)**						
	VSP-Ae global index (evaluation by the child)		64.38 (12.55)	70.30 (13.07)	70.16 (13.06)	0.029 *
	VSP-Ap global index (evaluation by the parents)		64.38 (12.35)	70.66 (10.51)	71.25 (10.61)	0.008 *
	Kidscreen global index (evaluation by the child)		68.43 (16.65)	73.26 (17.43)	71.91 (17.30)	0.293
	Kidscreen global index (evaluation by the parents)		62.52 (16.54)	70.16 (14.50)	71.33 (13.39)	0.003 *
**Neurocognitive assessment ^2^**						
	**WISC-IV ^2a^, mean (SD)**					
		VCI (Verbal comprehension index)	80.93 (9.46)	96.56 (11.75)	109.51 (12.74)	<0.001 *
		PRI (Perceptional reasoning index)	77.26 (8.99)	85.41 (9.24)	106.73 (9.94)	<0.001 *
		WMI (Working memory index)	74.93 (11.07)	88.85 (9.38)	101.15 (11.41)	<0.001 *
		PSI (Processing speed index)	80.15 (8.96)	88.49 (10.75)	102.90 (12.61)	<0.001 *
	**NEPSY-2 ^2b^, mean (SD)**					
		Planification score (Tower)	8.80 (3.04)	10.81 (2.33)	11.52 (2.43)	<0.001 *
		Mental flexibility score	6.50 (2.27)	7.76 (2.72)	9.48 (2.68)	<0.001 *
		Auditive attention score	8.61 (1.34)	8.45 (1.51)	9.66 (1.73)	<0.001 *
		Visual attention score	8.98 (2.73)	9.78 (3.47)	11.64 (3.03)	<0.001 *
	**Goodman-SDQ-parents ^3^, mean (SD)**					
		Emotional symptoms	4.05 (2.62)	3.64 (2.37)	3.15 (2.56)	0.136
		Behavioral problems	2.52 (1.99)	1.94 (1.93)	1.64 (1.73)	0.040 *
		Hyperactivity/Inattention	5.32 (2.45)	4.89 (2.56)	3.93 (2.74)	0.007 *
		Relationship problems with others	2.23 (1.92)	2.17 (1.98)	1.55 (1.77)	0.051
		Prosocial behaviors	8.59 (1.59)	8.61 (1.70)	8.89 (1.42)	0.412
		Total difficulty scores	14.11 (6.39)	12.64 (6.63)	10.27 (5.78)	0.002 *
	**Anxiety, mean (SD)**					
		Spielberg Index (STAIC) ^4^	35.30 (8.28)	31.90 (7.71)	32.25 (6.89)	0.035 *
**Impairment ^5^**						
	Language delay ^5a^, n (%)		30 (65.22)	16 (16.16)	1 (1.16)	<0.001 *
	Delay in visuospatial integration ^5b^, n (%)		17 (36.96)	18 (18.18)	0 (0.00)	<0.001 *
	Attention deficit disorder ^5c^, n (%)		15 (32.61)	19 (19.19)	2 (2.33)	<0.001 *
	Dysexecutive disorder ^5d^, n (%)		44 (95.65)	67 (67.68)	26 (30.23)	<0.001 *
	Ideomotor dyspraxia ^5e^, n (%)		14 (30.43)	10 (10.10)	0 (0.00)	<0.001 *

Note: Data are expressed in numbers (%) or mean (SD); SD: standard deviation. *P*: value of the difference between the groups with available data: *p* < 0.05 *: statistically significant difference. Abbreviations: GA: gestational age; WA: weeks of amenorrhea; SGA: small for gestational age; BPD: bronchopulmonary dysplasia; FAS: family affluence scale. **Cluster Classification**: Three neurobehavioral phenotype profiles chosen by the algorithm were defined. Cluster 1: Moderate Impaired NeuroBehavior (MODINB), Cluster 2: Minor Impaired NeuroBehavior (MINB), Cluster 3 UnImpaired NeuroBehavior (UINB). **Quality of life** (QoL) ^1^: VSP-Ae and VSP-Ap: Perceived life and perceived health of adolescents, quality of life questionnaires (assessments by both the child and parents, respectively). Kidscreen-infants and Kidscreen-parents: quality of life questionnaires (evaluation by children and parents, respectively). For both questionnaires, scores ranged from 0 to 100, with the higher the score, the better the quality of life. **Neurocognitive assessment**
^2^: WISC-IV ^2a^: Weschler Intelligence Scale for Children—4th edition; NEPSY-2 ^2b^: NEuroPSYchological assessment, 2nd edition. **Goodman-SDQ (Strengths and Difficulties Questionnaire)**
^3^: score correlated with the Child Behavior Checklist (CBCL) of Achenbach which included 25 items directed toward the parents. The questionnaire reflects a total assessment of the behavioral and emotional problems and defines five subcategories, each composed of five items: (**1**) Emotional disorder: mean: 0–3, slightly below mean: 4, high: 5–6, very high 7–10; (**2**) Behaviors: mean 0–2, slightly below average: 3, high: 4–5, very high 6–10; (**3**) Hyperactivity: mean 0–5, slightly below mean: 6–7, high: 7–8, very high 9–10; (**4**) Social relationship disorder: mean: 0–2, slightly below mean 3, high: 4, very high 5–10. The total difficulty score is the sum of the first four items. An increase of one point in the total difficulty score corresponds to an increase in the risk of developing a mental disorder. The categories were proposed in order to assess whether the child’s score is in the mean 0–13, slightly below the mean 14–16, high 17–19, or very high 20–40. (**5**) ”Prosocial” behavior is counted separately and varies in the opposite direction: mean 8–10, slightly below mean: 7, high: 6, very high 0–5. **STAIC ^4^:** State–Trait Anxiety Inventory for children. Each STAIC item (Spielberger) constitutes an assigned three-point rating scale with a value of 1, 2, and 3. The final score is obtained by adding the sub-scores for each item and ranges from 20 to 60. The normative data of the STAIC, by age and by sex, were established in the United States (Spielberger, 1973) on the basis of two samples of elementary school children: one sample consisted of 456 girls and 457 boys, in their fourth, fifth, and sixth years of primary school and the other consisted of 281 boys and 357 girls for the same scholastic years. The boy’s group: Anxiety Trait 36.7 SD: 6.32; Anxiety State 31.0 SD 5.71. The girl’s group: Anxiety Trait: 38.0 SD: 6; 68 Anxiety State: 30.7 6.01. Schematically: Standard anxiety state: Normal: 20–40, High Anxiety: higher than 40. **Impairment ^5^:** A specific cognitive impairment was considered in the GPQoL study if at least one of the five neuropsychological disorders (classification of mental illnesses according to Diagnostic and Statistical Manual of Mental Disorders, fourth edition (DSM-IV)) was observed: A delay in language was considered if the verbal comprehension index (VCI) ^5a^ was <85, a delay in visuospatial integration was considered if the perceptual reasoning index (PRI) was <85 and Rey figure ≤ 10th percentile, in copy mode ^5b^, an attention deficit disorder was considered if the auditory attention score was <8 and/or the visual attention score was <8, and if the processing speed index (PSI) ^5c^ was <85, dysexecutive disorders were considered if a working memory index (WMI) was <85 and/or a planning score was <8 and/or a mental flexibility score was <8 and/or inhibition score ^5d^ < 10th; an ideomotor dyspraxia was considered if the Touwen assessment test = complex coordination disorder with anomaly of movement planning and (PRI was <85 and/or IVT was <85) ^5e^.

**Table 2 children-08-00943-t002:** Quality of life comparisons of studied (*n* = 178) between cluster and reference populations.

		**Quality of Life between Study Population vs. Reference Population**									**Quality of Life between Three Clusters**								**Quality of Life between Cluster 3 vs. Reference Population**					
		**Study Population** **(*n* = 178)**		**Reference Population ^a^**	**Difference**		** *p* **	**Effect Size ^3^ (Ranking)**			**Cluster 1 ^b^** **(*n* = 32)**		**Cluster 2 ^c^** **(*n* = 79)**		**Cluster 3 ^d^** **(*n* = 67)**		** *p* **		**Reference Population**	**Difference**		** *p* **	**Effect Size ^3^ (Ranking)**	
		**Mean**	**SD**	**Expected Mean**	**Mean**	**SD**					**Mean**	**SD**	**Mean**	**SD**	**Mean**	**SD**			**Expected Mean**	**Mean**	**SD**			
**VSP-A ENFANT ^1^**																								
	Vitality	77.50	19.90	82.46	−4.95	19.96	0.001 *	−0.25			77.03	20.90	77.22	19.77	78.06		0.958		82.31	−4.25	19.77	0.083	−0.22	
	General well- being	72.61	17.93	78.38	−5.77	18.02	<0.001 *	−0.32			68.15	17.56	74.33	17.23	72.72	18.79	0.259		78.42	−5.70	18.73	0.015 *	−0.3	3
	Relationship with friends	46.89	28.11	58.91	−12.02	28.17	<0.001 *	−0.43	2		43.29	24.81	46.70	28.66	48.82	29.14	0.659		58.89	−10.07	29.37	0.006 *	−0.34	1
	Leisure	62.86	20.52	69.63	−6.77	20.57	<0.001 *	−0.33	3		60.16	18.21	63.96	20.81	62.85	21.37	0.679		69.39	−6.54	21	0.013 *	−0.31	2
	Relationship with family	74.10	19.05	73.19	0.91	19.10	0.528	0.05			74.38	17.33	74.97	18.55	72.94	20.58	0.813		73.00	−0.06	20.77	0.981	0	
	School work	76.19	23.50	82.08	−5.88	23.45	0.001 *	−0.25			60.94	24.75	78.8	21.87	80.41	22.11	<0.001 *		82.16	−1.75	21.98	0.518	−0.08	
	Self esteem	74.20	21.33	84.61	−10.41	21.42	<0.001 *	−0.49	1		63.54	22.71	74.74	21.39	78.67	19.00	0.004 *		84.49	−5.82	19.25	0.016 *	−0.3	
	Total index	69.19	13.56	75.59	−6.40	13.65	<0.001 *	−0.47			63.93	11.5	70.1	13.35	70.64	14.27	0.050		75.51	−4.87	14.31	0.007 *	−0.34	
**VSP-PARENTS ^1^**																								
	Vitality	69.92	16.15	77.38	−7.46	16.26	<0.001 *	−0.46			65.23	16.43	69.86	16.14	72.23	15.77	0.131		77.36	−5.13	15.66	0.009 *	−0.33	3
	Psychological well-being	70.19	20.43	81.34	−11.15	20.50	<0.001 *	−0.54	1		70.23	25.58	70.46	19.08	69.85	19.53	0.984		81.19	−11.34	19.57	<0.001 *	−0.58	1
	Relationship with friends	59.18	20.04	64.53	−5.36	20.15	<0.001 *	−0.27			53.45	20.28	59.49	21.28	61.54	18.08	0.169		64.71	−3.17	18.43	0.163	−0.17	
	Leisure	52.41	18.32	57.01	−4.60	18.30	0.001 *	−0.25			50.13	17.53	54.11	19.03	51.49	17.91	0.512		57.85	−5.36	17.85	0.017 *	−0.3	
	Relationship with family	77.21	13.65	78.62	−1.41	13.69	0.172	−0.10			75.91	16.08	78.80	13.63	75.96	12.39	0.386		78.51	−2.55	12.32	0.095	−0.21	
	Physical well being	75.78	16.18	78.60	−2.82	16.19	0.021 *	−0.17			74.28	16.32	76.58	17.03	75.56	15.26	0.789		78.38	−2.82	15	0.129	−0.19	
	Relationship with teacher	73.17	18.46	75.16	−1.99	18.67	0.157	−0.11			70.31	19.28	74.26	17.05	73.26	19.76	0.596		75.32	−2.06	19.78	0.398	−0.1	
	School work	69.66	20.14	79.80	−10.14	20.18	<0.001 *	−0.50	2		60.94	22.17	67.41	18.17	76.49	19.40	0.001 *		79.92	−3.43	19.67	0.158	−0.17	
	Self esteem	78.93	27.33	88.44	−9.50	27.43	<0.001 *	−0.35			62.11	32.14	81.17	24.59	84.33	25.13	<0.001 *		88.09	−3.76	25.71	0.236	−0.15	
	General well- being	72.72	15.81	80.12	−7.41	15.89	<0.001 *	−0.47	3		72.19	18.24	73.20	15.28	72.40	15.42	0.935		79.94	−7.54	15.38	<0.001 *	−0.49	2
	Total Index	69.61	11.17	75.96	−6.36	11.23	<0.001 *	−0.57			64.73	12.38	70.24	10.59	71.19	10.75	0.021 *		75.89	−4.69	10.81	0.001 *	−0.43	
**KIDSCREEN-ENFANT ^2^**																								
	Total index	72.23	17.63	76.87	−4.65	17.74	0.001 *	−0.26			67.39	17.7	73.82	17.26	72.66	17.89	0.213		76.78	−4.12	17.92	0.064	−0.23	
**KIDSCREEN-PARENTS ^2^**																								
	Total index	69.61	14.83	71.84	−2.23	14.83	0.046 *	−0.15			64.91	16.16	70.01	14.94	71.37	13.74	0.121		71.84	−0.47	13.74	0.781	−0.03	

Note: ^a^ Reference population: French reference population from the European database of 2003 which included a French sample (*n* = 989) obtained by randomized telephone calls (CATI method: Computer-Assisted Telephone Interview; RDD: Random Digital Dialing). ^1^ VSP-Ae and VSP-Ap: Perceived Life and Health of Adolescents, quality of life questionnaires (assessments obtained by the child and by the parents’ perspective of their child). The scores vary between 0 and 100. A high score indicates a better quality of life. ^2^ Kidscreen-infants and Kidscreen-parents Index-10 abridged version: quality of life questionnaires (evaluation by children and by parents’ perspective of their child). Scores range from 0 to 100. The higher the score, the better the quality of life. *p* < 0.05 *: statistically significant difference. ^3^ Effect size: Ranked by decrease in quality of life in each area for VSP-Ae and VSP-Ap. Standardized size effect was obtained by dividing the average difference by the standard deviation. 1, 2, and 3 and indicate the top 3 variables in the quality of life areas most affected. Cluster classifications: a classification in three neurobehavioral phenotype profiles was chosen by the algorithm. ^b^ Cluster 1: moderate impairment neurobehavior (ModINBP), ^c^ Cluster 2: minor impairment neurobehavior (MinINB), ^d^ Cluster 3: UnImpaired NeuroBehavior (UINB).

## Data Availability

The datasets that were generated and/or analyzed during the current study are not publicly available due to the data belonging to the Assistance Publique Hopitaux de Marseille. However, datasets are available from the sponsor (promotion.interne@ap-hm.fr) on reasonable request and after signing a contract pertaining to the provision of data and/or results.

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
