# Peer review of "The Neurobehavioral Phenotype of School-Aged, Very Prematurely Born Children with No Serious Neurological Sequelae: A Quality of Life Predictor"

_children, 2021, doi:10.3390/children8110943_

Round 1

Reviewer 1 Report

Congratulations on your nice work. However some revisions are necessary 

Author Response

Authors’ Response to Reviewers:

Dear reviewers,

Thank you for your letter and for the reviewers’ comments on our manuscript entitled “The neurobehavioral phenotype of school-aged, very prematurely born children with no serious neurological sequelae:  A quality of life predictor”. All of these comments were very helpful for revising and improving our paper. We have studied these comments carefully and have made corresponding corrections that we hope will meet with your approval. The changes in the revised manuscript are marked in red. The responses to the reviewers’ comments are provided below.

We would like to express our great appreciation to you and the reviewers for the comments on our paper.

Kind regards,

Reviewer 2 Report

The authors identified neurobehavioral subgroups in a highly preterm population who shared similar profiles on the measurements of intelligence and executive functions, and determined distribution of anxiety and behavioral disorders assessment in each group, and finally correlated with QoL measurements. This is a fascinating study. The paper is well written, and the rationale is very clear. However, I have minor comments that will improve understanding of the manuscript.

Please give the full form for the term before introducing the abbreviation. For example, it took me a moment to understand what the author meant by “VP” in line 59.

What is the GPQoL study? line 103

Figure 1 – how the author arrived to N=231 in the flow chart after n =178

The standard deviation is too high for components of Goodman-SDQ-parents. Please explain.

 How p-values in Table 1 were calculated. Is it between cluster 1 vs. cluster 2 vs. cluster 3?  t-test should not be used for more than two groups. ANOVA will be an appropriate test for comparing 3 groups.

What does the author mean by connecting lines in Figures 2 and 3? Line plots are misleading. It seems the author is using the relationship between the mean scores of VCI, PRI, WMI, and PSI. If the author wants to show the mean score of VCI, PRI, WMI, and PSI for each cluster, a histogram or box plot with standard deviation bars would be appropriate.

The author didn’t mention the interpretation and significance of effect size.

This is not exploratory study, and please provide p-values corrected for multiple tests. Also, there could be confounding factors like clinical comorbid condition, parent's age, access to health care, hemoglobin (anemia reported to be associated with cognitive decline), sex. Did the author adjust for these covariates/confounders?

Author Response

Authors’ Response to Reviewers:

Dear reviewers,

Thank you for your letter and for the reviewers’ comments on our manuscript entitled “The neurobehavioral phenotype of school-aged, very prematurely born children with no serious neurological sequelae:  A quality of life predictor”. All of these comments were very helpful for revising and improving our paper. We have studied these comments carefully and have made corresponding corrections that we hope will meet with your approval. The changes in the revised manuscript are marked in red. The responses to the reviewers’ comments are provided below.

We would like to express our great appreciation to you and the reviewers for the comments on our paper.

Kind regards,

Question: 

  1. Please give the full form for the term before introducing the abbreviation. For example, it took me a moment to understand what the author meant by “VP” in line 59.

Thank you for your attention, the correction has been made.

  1. What is the GPQoL study? line 103

See Method section, please. Line 108 (study design and population). Thank you

  1. Figure 1 – how the author arrived to N=231 in the flow chart after n =178

Thank you for your correction, Figure 1 has been changed to better understand the distribution:

  1. The standard deviation is too high for components of Goodman-SDQ-parents. Please explain.

Thank you for your pertinent comment. It is the amount of the first 5 items and for total SDQ (total difficulty score) it is normal to have an increase of SD.

  1. How p-values in Table 1 were calculated. Is it between cluster 1 vs. cluster 2 vs. cluster 3?  t-test should not be used for more than two groups. ANOVA will be an appropriate test for comparing 3 groups.

We confirm that ANOVA has been used, see last line paragraph « New classification of cognitive/behavioral impairments »: All data were then compared according to the three established neurobehavioral profiles for qualitative data, the χ2-test was used when valid (otherwise the Fisher-test was used). For quantitative data, the analysis of variance was used when valid (otherwise the Kruskal-Wallis-test was used).

  1. What does the author mean by connecting lines in Figures 2 and 3? Line plots are misleading. It seems the author is using the relationship between the mean scores of VCI, PRI, WMI, and PSI. If the author wants to show the mean score of VCI, PRI, WMI, and PSI for each cluster, a histogram or box plot with standard deviation bars would be appropriate.

Thank you, indeed it is not an appropriate form. We hope that it will be convenient for you.

  1. The author didn’t mention the interpretation and significance of effect size.

Thank you, we have clarified the purpose of effect size : As suggested by Cohen, d statistic lower than 0.2 represent a negligible difference even it is statistically significant while d statistic equals to 0.2, 0.5, and 0.8 represent a small, medium and large difference, respectively.

  1. This is not exploratory study, and please provide p-values corrected for multiple tests. Also, there could be confounding factors like clinical comorbid condition, parent's age, access to health care, hemoglobin (anemia reported to be associated with cognitive decline), sex. Did the author adjust for these covariates/confounders?

This is an exploratory study from a GPQoL cohort (RCT). In this sense, we have moderated our comments on the prediction data by trying to build models or clusters of premature children. Thank you.

Reviewer 3 Report

Line 59/314 - Acronym "VP"- Should not be used as it has not been mentioned before 

Line 111-"(28<SA)" looks a typo - may be GA <28 weeks ?

Line 202- 8.46(+ 0.75) , should mention "years"

Page 7/19-Cluster 1: Impairmed----> Impaired 

 Page 8/19- 3) Hyperreactivity ---high87-   -------->high 7-8 

Page 8/19- In dysexecutive disorders - Please delete "There was "

Line 235-6, 334: -1DS needs to be changed to -1SD 

Figure 4- In notes : "0-.5" should be 0-5.

Line 321 -"Both" can be deleted 

Line 391 -"Human warmth" has been used which does not convey the meaning , may need to elaborate or change wording like human relationships 

Line 404 -Drawing comparisons to autism may not be completely true as autism has a huge spectrum from very mild affected individuals to profound ones  

Author Response

Authors’ Response to Reviewers:

Dear reviewers,

Thank you for your letter and for the reviewers’ comments on our manuscript entitled “The neurobehavioral phenotype of school-aged, very prematurely born children with no serious neurological sequelae:  A quality of life predictor”. All of these comments were very helpful for revising and improving our paper. We have studied these comments carefully and have made corresponding corrections that we hope will meet with your approval. The changes in the revised manuscript are marked in red. The responses to the reviewers’ comments are provided below.

We would like to express our great appreciation to you and the reviewers for the comments on our paper.

Kind regards,

Question: 

Line 404 -Drawing comparisons to autism may not be completely true as autism has a huge spectrum from very mild affected individuals to profound ones  

Response:

We agree with you, the sentence have been modified. Thank you.

There are hypotheses that suggest this NBP of prematurity is due to cerebral hypo-connectivity, thus leading to diffuse structural anomalies that may sometimes be similar to autism spectrum disorders [27] (although there is a huge spectrum), but with a reduced severity.

Reviewer 4 Report

The study is well conducted and clearly discussed.

It gives an extensive view of the neurocognitive anomalies among the EPT group.

On the other hand, it also contributes to stress the factors that might be influenced in order to ameliorate the burden of disabilities.

Author Response

(The authors gave the same response as above.)

Round 2

Reviewer 1 Report

Congratulations on your work